# Glyoxalase System as a Therapeutic Target against Diabetic Retinopathy

**DOI:** 10.3390/antiox9111062

**Published:** 2020-10-30

**Authors:** Gemma Aragonès, Sheldon Rowan, Sarah G Francisco, Wenxin Yang, Jasper Weinberg, Allen Taylor, Eloy Bejarano

**Affiliations:** 1Laboratory for Nutrition and Vision Research, USDA Human Nutrition Research Center on Aging, Tufts University, Boston, MA 02155, USA; gemma.aragones@tufts.edu (G.A.); sheldon.rowan@tufts.edu (S.R.); sarah.francisco@tufts.edu (S.G.F.); wenxiny7@brandeis.edu (W.Y.); jasper.weinberg@tufts.edu (J.W.); 2Department of Ophthalmology, Tufts University School of Medicine, Boston, MA 02155, USA; 3Friedman School of Nutrition and Science Policy, Tufts University, Boston, MA 02155, USA; 4Universidad Cardenal Herrera-CEU, CEU Universities, 46115 Valencia, Spain

**Keywords:** diabetic retinopathy, oxidative stress, glycation, aging, glyoxalase

## Abstract

Hyperglycemia, a defining characteristic of diabetes, combined with oxidative stress, results in the formation of advanced glycation end products (AGEs). AGEs are toxic compounds that have adverse effects on many tissues including the retina and lens. AGEs promote the formation of reactive oxygen species (ROS), which, in turn, boost the production of AGEs, resulting in positive feedback loops, a vicious cycle that compromises tissue fitness. Oxidative stress and the accumulation of AGEs are etiologically associated with the pathogenesis of multiple diseases including diabetic retinopathy (DR). DR is a devastating microvascular complication of diabetes mellitus and the leading cause of blindness in working-age adults. The onset and development of DR is multifactorial. Lowering AGEs accumulation may represent a potential therapeutic approach to slow this sight-threatening diabetic complication. To set DR in a physiological context, in this review we first describe relations between oxidative stress, formation of AGEs, and aging in several tissues of the eye, each of which is associated with a major age-related eye pathology. We summarize mechanisms of AGEs generation and anti-AGEs detoxifying systems. We specifically feature the potential of the glyoxalase system in the retina in the prevention of AGEs-associated damage linked to DR. We provide a comparative analysis of glyoxalase activity in different tissues from wild-type mice, supporting a major role for the glyoxalase system in the detoxification of AGEs in the retina, and present the manipulation of this system as a therapeutic strategy to prevent the onset of DR.

## 1. Oxidative Stress is Related to Many Age-Related Eye Diseases

Age-related eye diseases such as cataract, age-related macular degeneration (AMD), glaucoma, and diabetic retinopathy (DR) are the main causes of progressive and irreversible vision loss worldwide [1]. Loss of vision caused by these diseases diminishes quality of life [2,3,4]. The World Health Organization (WHO) reported that in 2010, there were 285 million people visually impaired, of which 39 million were blind, and that by 2050 it is estimated that the number will triple, exacerbating the enormous personal and public health burdens [5,6]. 

The pathogenesis of age-related eye diseases is complex and depends on many factors, some of which remain to be identified. However, it is clear that oxidative stress and the resultant dysfunctional cellular moieties are pathoetiologic for the development of age-related eye diseases [7,8,9,10]. 

Oxidative stress is defined as the generation of excess reactive oxygen species (ROS) beyond the capacity of the biological systems that detoxify these reactive free radicals [11]. Examples of ROS include hydrogen peroxide (H_2_O_2_), superoxide (O_2_^•−^), and nitric oxide (NO). These imbalances lead to oxidative alterations of cellular macromolecular targets such as DNA, RNA, lipids, proteins, and carbohydrates, and eventually to dysfunction and degeneration of tissues [12,13,14]. In proteins, carbonyls are formed by the Fenton reaction of oxidants with lysine, arginine, proline, and threonine residues of the protein side chains [15]. Typical sequelae of oxidation are the formation of protein aggregates and impaired activity of many proteins, both structural and catalytic [16,17,18,19]. 

Although not the focus of this review, additional biological oxidation processes involve the formation of oxidized lipid metabolites such as hydroxynonenal, and the oxidation of lipids such as low-density lipoproteins in which both the protein and the lipids undergo oxidative changes that can cause cholesterol accumulation [20]. There is also oxidative damage to DNA resulting in several mutagenic lesions including 2-hydroxy adenine, 8-oxoadenine, 5-hydroxycytosine, cytosine glycol, thymine, and glycol [21]. 

A significant literature indicates that several eye tissues are particularly vulnerable to oxidative stress. Retinal photoreceptor cells and retinal ganglion cells have a large number of mitochondria, sustain high exposure to light and have a high rate of metabolism. As might be anticipated for a tissue in which light energy is transformed to chemical and then electrical impulses, which are chemically transmitted to the brain, retinal photoreceptor cells and retinal ganglion cells are susceptible to oxidative stress [8]. Outside the neurosensory retina, the retinal pigment epithelium (RPE) is also highly sensitive to photo-oxidative stress. The RPE is a single layer of cells located between photoreceptors and the choroid that plays key roles in the maintenance of photoreceptors. Oxidative stress is pathoetiologic in the RPE degeneration associated with the onset of AMD [9].

Oxidative stress is also pathoetiologic in other ocular tissues: (1) Damage to cell membrane fibers, lenticular proteins, photoreceptors, and DNA in most cells, (2) angiogenesis, endothelial dysfunction, and cell apoptosis, (3) loss of lens transparency by disrupting electrolyte balance homeostasis, and (4) increase of intraocular pressure and associated glaucoma [10]. Thus, it is not surprising that multiple ocular diseases have been linked to oxidative stress. These include retinitis pigmentosa, AMD, glaucoma, cataract, and others [12,22,23]. For example, intraocular pressure seems to increase with oxidative stress and accumulation of ROS in retinal ganglion cells [24]. In addition, glaucoma patients have diminished levels of antioxidant biomarkers such as vitamin E [23]. Crabb et al. using mass spectrometry, observed several oxidized proteins in drusen, extracellular deposits accumulated below the RPE on Bruch’s membrane, from human AMD samples [25]. The accumulation of drusen is considered to be indicative of early AMD. Oxidative stress also plays a pathologic role in the onset and progression of cataracts. Crystallins, the major gene products in the eye lens, see sulfhydryls transformed to disulfides in cross-linked and aggregated cataractogenic moieties [26].

Based upon these associations between oxidative stress and the risk for cataract, AMD, glaucoma, and DR, many studies have tried to elucidate whether the dietary intake of nutrients with anti-oxidative properties could prevent different age-related eye diseases [27,28,29,30]. In the age-related eye diseases double-blinded placebo controlled study (AREDS), it was demonstrated that consumers of fruit and vegetable-rich diets, and people who consume supplements of vitamins C and E, as well as zinc and lutein (all involved in antioxidant function) are protected against AMD [27,31].

## 2. Advanced Glycation End Products: A Special Case of Oxidative Stress Found in Aged Eye Tissues and Throughout the Body

Advanced glycation end products (AGEs) are oxidation products of particular interest for this review because they are associated with multiple diseases of aging, including DR [32,33,34,35,36,37]. 

Dietary sugars or dicarbonyls generated from carbohydrate metabolism can be highly reactive and transform many biomolecules and structures through a process called glycation. This non-enzymatic process is initiated with the Maillard reaction, in which a reversible Schiff base is formed between the carbonyl group of reducing sugars and free amino groups of proteins (Figure 1). These precursors undergo additional oxidations and rearrangements, resulting in the biogenesis of Amadori products. Depending on pH, Amadori products can rearrange to different types of dicarbonyls including 1,2-dicarbonyls such as methylglyoxal (MG) or 3-deoxyglucosone (3-DG), or 2,3-dicarbonyls such as 1-deoxyglucosone. The major glycating biologic reagent is MG, formed by the degradation of dihydroxyacetone phosphate and glyceraldehyde 3-phosphate, both glycolytic metabolites, as well as by the metabolism of threonine, the oxidation of ketone bodies, and upon degradation of glycated proteins [38]. Glyoxal and 3-DG are also highly reactive dicarbonyls formed during sugar metabolism [32,33]. These dicarbonyls are maintained at low levels under homeostatic conditions. Subsequent reactions result in methylglyoxal-derived hydroimidazolone 1 (MG-H1), Nε-carboxy-methyl-lysine (CML), Nε-carboxy-ethyl-lysine (CEL), pentosidine, glucosepane, and other types of AGEs [39] (Figure 1). As discussed later, AGE accumulation boosts the formation of ROS, resulting in increased production of AGEs. This vicious cycle of the oxidative formation of AGEs impacts cellular metabolism and contributes to hyperglycemia-induced tissue injury.

AGEs are accumulated throughout the body upon aging and particularly in diabetic patients. Three decades ago, different studies reported higher levels of different AGEs in diabetic tissues [40,41,42,43]. One of the best examples of glycation with aging is the extracellular matrix. Examination of interstitial collagen shows a gradual increase in AGEs upon aging [44] (Figure 2A). Another example of the relation between AGEs accumulation and aging occurs in eye tissues. Cataracts are perhaps the earliest example of pathobiology of AGEs in aged tissues [45]. Human lens crystallins become progressively yellow-brown pigmented with age as a result of the accumulation of Maillard products [46] (Figure 2B,C). The extensive modification of crystallins by glycation alters the dynamic state of crystallins, promotes their aggregation, and disrupts their chaperone function, contributing to cataractogenesis.

The deleterious effect of glycative damage is cell-type dependent and the molecular consequences of AGEs accumulation occur at different levels in eye tissues [47]. Regarding DR, the involvement of AGEs in the pathogenesis of the disease is complex. Glycation of the extracellular matrix results in decreased elasticity and increased vascular stiffness, leading to abnormal vascular function due to rigidity of the vessel wall. AGEs can also exert an indirect effect by binding to different cellular receptors in the plasma membrane, triggering intracellular pathways such as NF-κB activation via Ras-MAPK or RhoA/ROCK pathways [48]. As a result, changes in intracellular signaling cascades and cytopathological responses are triggered that include releases of pro-inflammatory cytokines and pro-angiogenic factors. This also leads to ROS generation, pericyte apoptosis, vascular inflammation, angiogenesis, changes in vasopermeability, and compromised blood–retinal barrier [47] (Figure 3). 

## 3. Role of Advanced Glycation End Products in the Pathogenesis of DR

Among the best studied pathologies related to oxidative stress are diabetes and DR, which occur in about 15% of those with long-lasting diabetes mellitus. DR is characterized by high levels of circulating sugars, high levels of oxidative stress, accumulation of AGEs, and microvascular damage [49]. DR is a leading cause of blindness in adults. During the early stage of DR, there is a loss of pericytes from capillaries, leading to the formation of acellular capillaries and retinal microaneurysms, along with thickening of the capillary basement membrane. Oxidative stress enhances damage to tight-junction complexes, causing vascular permeability, and blood–retinal barrier damage [50]. Together, these pathological changes result in irreversible damage to the blood–retinal barrier. Once the disease progresses to a late stage, neovascularization and bleeding can occur along with retinal detachment and macular edema, ultimately resulting in vision loss [51]. 

Several lines of evidence point to a relationship between oxidative stress and glycation-derived damage. Urinary 8-hydroxy-2’-deoxyguanosine, a marker for oxidative stress, was positively associated with glycated albumin levels in patients with type 2 diabetes, whereas improved glycemic control was associated with decreased levels of oxidative stress [52]. *In vitro* experiments showed that glycated human serum albumin protein promoted sustained ROS production in human endothelial cells. These findings support the hypotheses that there is a vicious cycle of oxidative stress, formation of AGEs, and production of more ROS. Furthermore, long-term oxidative stress induced by AGEs results in endothelial dysfunction which is associated with DR [53]. Indeed, oral administration of AGEs through a diet highly enriched in AGEs promoted oxidative stress, increasing inflammation and insulin resistance [54]. In addition, it has been shown that levels of H_2_O_2_ and O_2_^•−^ are increased in the retinas of diabetic rats, consistent with oxidative stress as a component of the pathobiology of DR [55,56]. Together, this experimental evidence indicates a relationship between elevated levels of ROS, AGEs, and DR.

Glycation is a critical biological process in the retina given that this ocular tissue has high metabolic activity whose activity depends on glucose demand. Such is the case in other neural tissues with the same embryological origin such as the cerebral cortex. The retina and cerebral cortex are exposed to comparable levels of blood glucose; however, the retina is more vulnerable to microvascular lesions derived from hyperglycemia. For example, in diabetic rats, intracellular concentrations of glucose increased significantly in the retina but not in the cerebral cortex, suggesting that a differential response of glucose uptake might contribute to the higher susceptibility of the retina to diabetes-induced microvascular complications [57].

The reasons for the high vulnerability of the retina to glycative stress remain unclear but some evidence indicates that the regulation of glucose uptake could be key. In the retina, the glucose delivery from systemic circulation occurs across retinal capillaries and the RPE. Glucose transport is mainly mediated by the sodium-independent glucose transporters (GLUTs). There are fourteen GLUTs, of which five are well characterized [58]. GLUT1 and GLUT3 are expressed in all cell types (including retina, lens, brain, vascular endothelium). GLUT2 is mainly expressed in the liver, kidney, intestine, and in beta cells of the pancreas. GLUT4 is found in the heart, adipose, and skeletal muscle. Transport of glucose via GLUT1 is insulin independent, thus GLUT1 is always “open”, allowing unimpeded transport of glucose. GLUT1 is expressed in retina and retinal capillaries, although the highest levels are found in the RPE. Thus, the RPE appears to be the major route for glucose delivery from the choriocapillaris to the neural retina [59]. Changes in the levels of this glucose transporter have been associated with the development of DR [60]. Levels of GLUT1 decreased in the retina of diabetic-induced rats; however, the expression of GLUT1 in the RPE was not affected by diabetes, suggesting that the trans-epithelial transport of glucose was not compromised [61,62,63]. The limited capacity of retinal endothelial cells to modulate glucose uptake makes them highly sensitive to the detrimental consequences of hyperglycemia in diabetes.

There are several plasma membrane proteins with the capacity to bind AGEs. The best studied receptor for AGEs in the context of DR is the receptor for advanced glycation end products (RAGE), also called AGER [64]. The binding of AGEs to RAGE is engaged in vital cellular processes such as inflammation, apoptosis, or proliferation and associated with the development of different human diseases. It is reported that RAGE-dependent signaling plays a major role in microvascular diabetic complications. In the eye, RAGE is expressed in multiple cells including pericytes, endothelial cells, microglia, Müller glia, and retinal pigmented epithelium cells, and expression of RAGE is increased under diabetic conditions [65,66]. This would appear to exacerbate the influx of glucose. Consistent with RAGE function, upon diabetes induction, *RAGE* knockout mice had reduced acellular capillary formation and showed less retinal vasopermeability, microglial activation, and Müller cell gliosis [67]. 

A vast amount of literature associates glycation and AGEs with the progression of DR. AGEs found in skin were shown to predict the risk of DR progression [68,69]. Intravenous administration of AGEs increased retinal vascular leakage *in vivo* by stimulating the expression of vascular endothelial growth factor (VEGF) and decreasing levels of the antiangiogenic pigment epithelium-derived factor (PEDF) [70].

Compounds that inhibit the formation of AGEs are reportedly effective against DR in preclinical settings [71]. An inhibitor of AGEs formation, aminoguanidine, significantly reduced serum AGEs and prevented the development of diabetes-induced basement membrane expansion in retinal capillaries of diabetic rats [72]. Aminoguanidine also reportedly reduced the number of acellular capillaries and abnormal microthrombus formations [73,74,75]. In addition, two different randomized, double-blind placebo-controlled trials reported that two different AGEs inhibitors, aminoguanidine and pimagedine, slow the progression of diabetic complications including DR [76,77]. Unfortunately, adverse side-effects of these AGEs inhibitors preclude their use in humans, and clinical trials for most of AGEs inhibitors have been discontinued. There is, therefore, a need to identify other means of lowering toxic AGEs.

## 4. Protective Mechanisms against Glycation-Derived Damage

### 4.1. Antioxidant Enzymes, Antioxidants, and Signaling 

Since the biogenesis of AGEs involves oxidative stress, it might be anticipated that an ability to scavenge ROS would provide a first line of defense. Such enzymatic capacities are provided by antioxidant enzymes such as catalases, H_2_O_2_-detoxifying enzymes, and superoxide dismutases, as well as glutathione recycling enzymes, glutathione reductases, and glutathione peroxidases [78]. There are also non-enzymatic antioxidants, the most prevalent and most potent of which are ascorbate and glutathione (GSH). In the eye, these are present at mM concentrations, many times the levels in the blood [79,80,81,82]. GSH provides a reducing cellular environment. However, it is also a critical component of the glyoxalase system. It also regulates the ubiquitin proteolytic pathway (discussed later in the review). The GSH/GSSG ratio establishes cellular redox status, with consequences for many metabolic processes. Thus, scavenging ROS might be a useful strategy to diminish the production of AGEs in our tissues.

In addition to ascorbate, other non-enzymatic ROS-scavengers, albeit less effective on a per molecule basis, include vitamin E, carotenoids, and flavonoids. Several natural compounds with antioxidant properties have attracted attention with regard to DR, including multiple polyphenols, some of which have anti-glycating potential. For example, administration of curcumin to streptozotocin-induced diabetic rats enhanced the antioxidant capacity in the retina [55] and there was a protective effect on the glycation and crosslinking of collagen [83]. In addition, flavonols such as quercetin, catechin, or kaempferol were shown to diminish AGEs formation [84,85,86]. Treatment of hepatic cells with a hydroxytyrosol-enriched extract from olive leaves reduced protein carbonylation and the formation of AGEs [87]. Several studies also showed attenuating effects for resveratrol on the production of AGEs or the RAGE receptor in cell culture, animal models, and human studies (reviewed in [88]). 

Among the pathobiological mechanisms by which AGEs enhance ROS production are perturbations in cellular signaling associated with the interaction between AGEs and RAGE. The production of free radicals by NADPH-oxidase and the mitochondrial electron transport system was shown to involve the AGEs–RAGE axis. The stimulation of the AGEs–RAGE signaling generates ROS by activating NADPH oxidases, augmenting the intracellular levels of H_2_O_2_, O_2_^•−^, and NO [89,90,91]. Interestingly, AGEs-induced upregulation of H_2_O_2_ production, along with mitochondrial dysfunction, resulted in apoptosis in ARPE-19, an RPE-derived cell line from normal eyes [92].

Several studies found that the upregulation of AGE-Receptor 1 (AGER1) inhibited the activity of NADPH oxidase, thereby weakening ROS production [93,94]. AGER1 is linked to sirtuin1 (SIRT1) [54,95]. In this context, deacetylation of NFκB by SIRT1 reduced the NFκB-mediated proinflammatory response [54]. However, long-term exposure to AGEs reduced AGER1 and SIRT1 expression, causing oxidative stress, inflammation, and insulin resistance in several tissues [54]. Another study showed that the dietary limitation of AGEs in type 2 diabetes mellitus patients diminished insulin resistance and increased expression of SIRT1 and AGER1 [94].

In sum, glycative stress contributes to the pathogenesis of different human diseases, including DR, through the interaction of oxidative stress and AGE accumulation.

### 4.2. Detoxifying Mechanisms against AGEs: Glyoxalase System, Aldehyde Dehydrogenases (ALDH), Aldoketoreductases (AKR), DJ-1/Park7, and Aldol Condensations

There are also specific protective pathways with the capacity to detoxify reactive dicarbonyls formed during sugar metabolism [96,97,98]. Once AGEs are formed, most are irreversible, so these protective mechanisms diminish the accumulation of AGEs in our tissues through the clearance of AGEs intermediates.

The primary mechanism for detoxifying these reactive dicarbonyls is the glyoxalase system. This converts highly reactive MG into far less reactive D-lactate [99] (Figure 3). This process involves the sequential activity of two enzymes, glyoxalase 1 (GLO1) and glyoxalase 2 (GLO2), and a catalytic amount of GSH. First, GSH reacts spontaneously with the aldehyde of the dicarbonyl to form a hemithioacetal adduct. Then, GLO1 catalyzes the formation of *S*-D-lactoylglutathione. In the second enzymatic step, GLO2 catalyzes the reaction of *S*-D-lactoylglutathione into D-lactate, regenerating GSH. It is important to note that GLO1 activity is proportional to GSH levels, and that its activity decreases if cellular cytosolic GSH is diminished, as upon oxidative stress [100]. Unfortunately, such diminution of GSH also compromises the function of the ubiquitin proteasome system, diminishing the cellular capacity to degrade AGEs and cope with glycation-derived damage [19]. There are additional substrates metabolized via this pathway, including glyoxal, phenylglyoxal, and hydroxypyruvaldehyde [101]. Of note, GLO1 is the rate-limiting enzyme, catalyzing the first detoxification step in the glyoxalase system, and its activity is required to prevent the accumulation of these reactive α-oxoaldehydes [102]. Thus, GLO1 has a key protective role against glycative stress-induced AGEs formation.

The structure of glyoxalase is informative. Briefly, the human GLO1 translation product contains 184 amino acids. It is a dimeric protein of molecular mass 42 kDa and contains one zinc ion per subunit. Its structure has two domains per subunit and there are two active sites per protein formed by amino acid residues from the apposed subunit such that the monomer is inactive. The human *GLO1* gene is diallelic (on 6p21.2) and encodes two similar subunits in heterozygotes that result in the dimeric holoenzyme. The two alleles, *GLO1* and *GLO2*, give two similar subunits and three dimers: allozymes GLO 1-1, GLO 1-2, and GLO 2-2. The only amino acid difference between the expression products of the two *GLO1* alleles is at position 111 (Ala^111^ or Glu^111^). Some studies showed association of this polymorphism with the variations of diabetes and its vascular complications [103]. In addition, in stage 5 renal failure patients on hemodialysis, the Glu111Glu homozygote was associated with increased prevalence of cardiovascular disease and peripheral vascular disease [104].

Transcriptional regulation of *GLO1* is only partially understood, but it is known that the *GLO1* sequence contains multiple regulatory elements. These include a metal-response element (MRE), insulin-response element (IRE), and early gene 2 factor isoform (E2F4) and activating enhancer-binding protein 2α (AP-2α) binding sites [98,105]. IRE and MRE functionalities were validated in reporter assays where insulin and zinc chloride exposure produced a 2-fold increased transcriptional response [105]. Similar functional activities were shown for E2F and AP-2α [106,107]. As discussed later, an antioxidant-response element (ARE) in exon 1 of *Glo1* enhances its transcription through the nuclear factor erythroid 2-related factor 2 (NRF2) oxidative stress-responsive transcriptional system [108].

Several systems appear to operate in the absence of glyoxalase activity, albeit their biological importance is largely unexplored. Other alternative routes for detoxification of dicarbonyls are aldehyde dehydrogenases (ALDHs), aldo-keto reductases (AKRs), the Parkinson-associated protein DJ-1, and scavenging by acetoacetate to form 3-hydroxyhexane-2,5-dione (3-HHD) [97,109].

AKRs are a large protein superfamily that is responsible for the reduction of aldehydes and ketones into primary and secondary alcohols. AKRs metabolize MG to hydroxyacetone or lactaldehyde. Transgenic expression of both human (*AKR7A2*) and mouse (*Akr7a5*) AKR in hamster fibroblasts cells protected against MG-induced cytotoxicity, suggesting that AKRs are able to detoxify MG and decrease AGE levels [110,111,112]. The role of AKR1B3 has also been studied in the hearts of induced diabetic *Akr1b3*-null mice, and it was observed that these mice had increased levels of MG and AGEs [112]. Human studies showed that aldose reductase activity (AKR1B1) contributed to the detoxification of MG in tissues where the protein was overexpressed and the levels of GSH were low [113]. An increase of AKR1B3 activity was observed in Schwann cells lacking glyoxalase activity, suggesting a compensatory relationship between these systems [114]. The theme of compensatory relationships between GLO1 activity and levels of various glycolytic metabolites will be noted in other systems, below.

ALDHs also metabolize MG, by oxidation to pyruvate. MG treatment induced ALDH expression in wild-type mouse Schwann cells [114]. Additionally, in both zebrafish and mouse models lacking glyoxalase activity, ALDH activity was induced, apparently as a compensatory mechanism [115,116].

3-deoxyglucosone (3-DG) is another highly reactive dicarbonyl formed during sugar metabolism. Although the physiological relevance of 3-deoxyglucosone (3-DG) in DR remains to be established, the 3-DG metabolite formed by ALDH1A1, 2-keto-3-deoxygluconic acid, was elevated in plasma and erythrocytes of diabetic patients [117]. High ALDH1A1 activity was also found in lung, testis and liver but is negligible in other tissues [118]. Interestingly, aldehyde detoxifying capacity is found in retina, however ALDHs were found to be downregulated in diabetic conditions [119].

Another protein with anti-glycation activity is DJ-1, also known as Parkinson’s disease protein 7 (PARK7) [120,121]. DJ-1 was shown to have glyoxalase activity *in vitro*, converting MG into lactic acid, in the absence of GSH, and preventing MG-induced tissue damage in *C. elegans* [122]. In addition, DJ-1 was shown to repair methylglyoxal- and glyoxal-glycated proteins in vitro. This deglycase activity removes early-stage MG adducts from protein side chains and prevents the formation of irreversible AGEs [123]. However, no contribution of DJ-1 to MG accumulation was observed in DJ-1 knockdown Drosophila cells and DJ-1 knockout flies [124]. A recent finding shows that DJ-1 may function as a relevant DNA deglycase [125] and recent studies observed a similar deglycase function for DJ-1 on DNA-wrapped histone proteins [126,127].

It was also reported that the ketone body acetoacetate decreased MG via a non-enzymatic conversion during diabetic and dietary ketosis [128]. Indeed, acetoacetate was able to scavenge endogenous MG in a non-enzymatic aldol reaction [129]. This has great importance since physiological ketosis produces high levels of acetoacetate and this condition may prevent diabetes progression [130].

### 4.3. Proteolytic Pathways: The Last Line of Defense against Glycation-Derived Damage

Which other mechanisms limit the accumulation of AGEs-modified proteins? Although AGEs are irreversible adducts and cross-links in our tissues, these can be removed through different proteolytic capacities. AGEs are substrates of intracellular protein degradation pathways and two major proteolytic capacities are suggested to contribute to the clearance of AGEs: the ubiquitin proteasome system (UPS) and autophagy [96,131,132,133,134,135] (Figure 3).

As for other cargos, AGEs-modified proteins were shown to be ubiquitinated [132]. Ubiquitin is a 76 amino acid protein that when conjugated to a protein substrate can facilitate degradation of that substrate by the proteasome. Obsolete or damaged proteins are tagged with ubiquitin and these ubiquitinated substrates are degraded by the proteasome. The ubiquitin proteolytic system operates mainly on soluble substrates. Several lines of evidence point to a relevant role for the UPS in the clearance of AGEs. Pharmacological proteasomal inhibition boosts the accumulation of AGEs *in vitro* in RPE-derived cells [132]. However, excessive glycative stress decreases proteasomal capacity via the formation of intermolecular crosslinks that, consequently, lead to accelerated accumulation of AGEs [136,137,138].

Autophagy targets cargos for degradation and can operate on insoluble substrates, including organelles such as mitochondria. Autophagy requires macromolecular assemblies and organelles to identify, sequester, and eventually degrade substrates via the lysosome. Both proteolytic routes, autophagy and UPS, are functionally cooperative and a deficiency of one of these pathways triggers the upregulation of the other [139,140]. Several reports support a vital role for autophagy in the removal of AGEs. Pharmacological blockade of autophagy *in vitro* induced higher accumulation of AGEs in RPE cells [132]. Accumulation of AGEs was observed in kidney tubules of diabetic autophagy-deficient mice [131]. Importantly, mice lacking autophagy in the RPE showed increased levels of oxidized and glycated proteins and were predisposed to develop AMD-phenotypes and retinal degeneration [133].

In sum, a significant literature supports a critical role for the UPS and autophagy in maintaining non-toxic, homeostatic levels of AGEs. Unfortunately, the function of both proteolytic pathways declines with extensive glycative stress and upon aging in many tissues [141], resulting in intracellular accumulation of protein aggregates (also glycated conjugates) and dysfunctional organelles in aged tissues [132,142,143]. We propose that deficits of these pathways in diabetic conditions could contribute to the accumulation and deposition of AGE-modified proteins in the retina, thereby contributing to DR. To date, there is scarce information about how these proteolytic pathways remove AGEs. This thwarts development of strategies to lower AGEs accumulation by boosting proteolytic capacities (Figure 4).

### 4.4. Protective Role of NRF2 against Glycation-Derived Damage and Modulation of GLO1

NRF2 is an essential transcription factor for genes encoding a number of detoxification enzymes that contain one or more antioxidant response elements (AREs) in their regulatory regions. Examples of genes with AREs include glutathione S-transferases, UDP-glucuronosyltransferases, aldo-keto reductases, and NAD(P)H:quinone oxidoreductase 1. NRF2 acts indirectly by upregulating genes that metabolize and excrete the causative agents and byproducts of oxidative stress [144].

NRF2 is highly conserved from mammalian species to chicken and zebrafish, particularly within six regions designated the Neh1–6 domains [145]. The Neh1 domain contains the cnc-bZIP region, which dictates dimerization partners and confers DNA binding specificity. The Neh4 and Neh5 domains act cooperatively to bind the coactivator CREB-binding protein, thereby activating transcription [146].

Under physiological conditions, NRF2 is in the cytoplasm in a complex with KEAP1, a substrate adaptor protein for the cullin-3-dependent E3 ubiquitin ligase complex. This directs NRF2 for degradation by the 26S proteasome [147,148]. Under oxidative stress, the complex dissociates and NRF2 translocates into the nucleus and upregulates several antioxidant genes, including those related to MG metabolism, as well as genes required for glutathione synthesis [149,150,151].

Several studies investigated the effect of NRF2 activators on MG and AGEs formation and deposition. Hepatic, brain, heart, kidney, and lung Glo1 mRNA and protein were decreased in *Nrf2*-knockout mice [108]. Compounds that increase *GLO1* expression and activity, such as sulforaphane or trans-resveratrol, decreased cellular and extracellular concentrations of MG and MG-derived protein adducts [152,153,154,155,156]. In addition, the binding of NRF2 to the *Glo1*-ARE increases expression of *Glo1*; however, the inflammatory activation of NF-κB (nuclear factor κB) by NRF2 could diminish *Glo1* expression [157]. Of note, NF-kB directly can also downregulate GLO1 activity, and the inhibition thus takes place at both functional and transcriptional levels [158].

Other transcription factors can modulate the expression of *Glo1*. Under hypoxia conditions, *Glo1* expression is inversely regulated by HIF1α (hypoxia-inducible factor 1α), an important physiological driver of dicarbonyl stress [159]. In addition, *Glo1* is acetylated and probably deacetylated by cytosolic sirtuin-2 [160,161], and its expression may be decreased by activation of the RAGE [127,162].

Several additional observations regarding the above control pathways and GLO1 are noteworthy. GLO1 downregulation is linked to activation of the RAGE receptor that is involved in pro-inflammatory signaling and the development of vascular complications of diabetes [127,162]. Upon glycative stress, there is increased GLO1 ubiquitination and degradation in high glucose concentration media *in vitro* [163]. Downregulation of NRF2 signaling is linked to decreased expression of *Glo1* and reduction of NRF2-antioxidant pathway signaling is associated with inflammation [108,157]. Nevertheless, even in Nrf2-knockout mice, GLO1 protein is still present in the retina and can even be moderately increased through dietary treatment, suggesting that NRF2 is just one of many different regulatory nodes for *Glo1*, at least in this specific ocular tissue [164]. The molecular mechanisms behind the increased retinal GLO1 stability in mice lacking NRF2 remain unknown.

## 5. The Use of Glyoxalase 1 in Animal Models

As previously explained, AGEs precursors are mainly detoxified by the glyoxalase system and, given the impact of AGEs on age-related pathologies, there is a need to develop strategies to counteract the accumulation of these toxic compounds. In the last few years, the manipulation of GLO1 activity in both animals and cell lines has demonstrated the causal involvement of MG and AGEs in several diseases.

In *Glo1* knockdown cell culture and animal models, an increase in free MG and toxic AGEs was described, but compensatory mechanisms were also observed [165,166]. Giacco et al. showed that in non-diabetic mice, knockdown of *Glo1* increased MG modification of glomerular proteins and oxidative stress to diabetic levels, causing alterations in kidney morphology indistinguishable from those caused by diabetes [167]. Surprisingly, in cell culture and animal models where *Glo1* was knocked out using CRISPR-Cas9 technology, there was no observed increase in MG accumulation [114,116,126,168]. In addition, these healthy *Glo1* knockout animal models did not present defects during development, while genetic deletion of *Glo1* was shown to be embryonically lethal in mice [169]. MGO treatment of cells lacking *Glo1* also revealed a decreased median lethal concentration for exogenous MG [114,126], indicating compensatory mechanisms for the loss of *Glo1*. Indeed, the authors demonstrated in this study that the deglycase protein DJ-1 may play a role in limiting the accumulation of MG-H1 on chromatin in cells lacking *Glo1*. Conversely, *Drosophila melanogaster* and *Danio rerio Glo1* knockout models showed an increase in tissue MG [115,170]. Moraru et al. observed increased MG levels, lipid accumulation in tissues, increased blood glucose, and decreased insulin sensitivity in *Drosophila melanogaster Glo1* knockout [170]. Consistent with these observations, Lodd et al. found under high nutrient intake increased MG levels driving insulin resistance and hyperglycemia in *Danio rerio Glo1* knockout [115].

Other studies have used the gene overexpression of *Glo1* in order to evaluate the biological impact of the glyoxalase system. The overexpression of *Glo1* reduced basal MG concentration, prevented mitochondrial protein modification, and enhanced lifespan in worms [171]. Similarly, *Glo1* overexpression reduced baseline MG concentration in the brain of mice [172]. In diabetic mice, *Glo1* overexpression also prevented diabetes-induced increases in MG modification of glomerular proteins, reduced oxidative stress, and prevented the development of diabetic kidney pathology, despite unchanged levels of hyperglycemia [165]. In agreement with mouse studies, rat models overexpressing human *GLO1* led decreased MG levels, less AGEs formation, and reduced renal and endothelial dysfunction in response to induced diabetes compared with wild-type littermates [173,174,175]. Recently, GLO1 and aldose reductase were found to be upregulated in patients protected against diabetic nephropathy [176], suggesting that the manipulation of the glyoxalase system could be a potential therapeutic strategy to prevent the onset of AGEs-related diseases.

In addition, diabetes and its microvascular complications (nephropathy, retinopathy, and neuropathy), are associated with elevated levels of MG and reduced levels of GLO1 expression and activity. Some studies showed that overexpression of *Glo1* in transgenic rats and mice could prevent the development of nephropathy, retinopathy, and neuropathy [167,174,175]. In vitro studies showed increased levels of MG and decreased GLO1 activity in endothelial cells when cultured in high glucose concentration media [102,177,178]. In contrast, overexpression of *Glo1* in endothelial cells under the same conditions prevented increased formation of AGEs [102]. Additionally, microvascular complications of diabetes linked to high glucose concentration in retinal pericytes were prevented by overexpression of *Glo1* [179]. In animal models, *Glo1* expression was decreased in the kidney of obese (db/db) diabetic mice, in the kidney and the sciatic nerve of streptozotocin-induced diabetic mice, in the kidney and liver of streptozotocin-induced diabetic Sprague–Dawley rats, in streptozotocin-induced diabetic Wistar rats, and in streptozotocin-induced rats overexpressing the renin-angiotensin system in extra-renal tissues [180,181,182,183,184]. By contrast, GLO1 activity was increased in red blood cells of streptozotocin-induced diabetic C57BL/6 mice, compared to non-diabetic controls [185] and GLO1 activity was increased in the red blood cells of patients with type 1 diabetes and type 2 diabetes, compared to healthy control subjects [186]. Patients with diabetes and microvascular complications had significantly higher GLO1 activity in red blood cells compared to patients without complications. These findings suggest a compensatory increase in GLO1 activity in response to elevated MG concentration [186] and that the response of GLO1 under diabetic conditions may be cell- and tissue-dependent.

### 5.1. The Decline of Glyoxalase 1 Activity with Age

Aging is characterized by a reduction in the functional properties of cells, tissues, and whole organs, starting with the impairment of major cellular homeostatic processes, including mitochondrial function, proteostasis, and stress-scavenging systems [166,187,188].

GLO1 expression and activity are modified upon aging and in age-related diseases including diabetes and its microvascular complications such as DR. Dicarbonyl stress contributes to aging through the age-related decline in GLO1 activity [171,186,189,190,191,192,193,194,195,196]. The first study that showed a causal link between aging and GLO1 was presented by Morcos et al. [171]. They found a marked decline of GLO1 expression and activity in *C. elegans* with age. Moreover, overexpression of *Glo1* was associated with prolonged lifespan, whereas *Glo1* silencing decreased lifespan, demonstrating that a decrease in GLO1 activity increases mitochondrial ROS production, thereby limiting lifespan [171]. In mice, Sharma–Luthra et al. found tissue specific differences. They showed that GLO1 activity diminished in liver and spleen with age, but increased in kidneys to maximum levels at 24 months [191]. GLO1 activity in rat tissues was decreased with age, as well as by hypoxia in young rats [192]. In humans, several studies have been conducted to investigate the impact of aging on GLO1. Most of them reported a decline of GLO1 activity in multiple tissues, such as arterial tissues, lens, brain, and red blood cells, with age [186,193,194,195,196].

### 5.2. Glyoxalase 1 Activity in Ocular Tissues

In spite of the vast literature on non-ocular tissues, there is scarce information about the role of the glyoxalase system in ocular tissues. However, increasing evidence points to a link between alteration of the glyoxalase system and the development of DR. A polymorphism that alters *GLO1* promoter activity has been linked to retinopathy in type 2 diabetic patients [197]. Expression of *GLO1* and *GLO2* are downregulated in patients with DR, indicating that a failure of this detoxifying system in humans may be involved in retinopathy [198,199]. Glyoxalase activity is reported *in vitro* to promote pericyte survival under hyperglycemic conditions [179] and retinal extracts from a mouse model protected from hyperglycemia-evoked vasoregression showed higher GLO1 activity [200]. Interestingly, a transgenic rat model overexpressing Glo1 inhibits retinal AGE formation and prevents DR lesions [174]. In sum, upregulation of GLO1 appears to reduce retinal AGEs in diabetic rats and ameliorate AGEs-related pathologies.

In order to explore the role of the glyoxalase system in the eye, we dissected retina, RPE/choroid, and lens, along with other non-ocular tissues from 2-month-old wild-type C57BL/6J mice. We quantified the GLO1 activity in cytosolic extracts of tissues, as previously reported [201]. As expected, we found activity in all tissues analyzed and the relative order of GLO1 activity was retina > liver > kidney > brain > heart > RPE/choroid > lens (Figure 5A). These results are consistent with previous publications in non-ocular tissues [116,184]. Regarding ocular tissues, we observed the highest activity of GLO1 in the retina of mice compared to the lens or RPE/choroid (Figure 5B). Glyoxalase activity is clearly tissue-dependent and retinal activity was about 9-fold and 13-fold greater than in the RPE/choroid and lens, respectively (Figure 5C,D). Clearly, there is even region-specific localization within tissues. This finding is highly relevant because avoiding the formation and accumulation of AGEs is especially important in highly differentiated tissues such as the retina or lens where the glycation damage cannot be diluted by cellular division [96,132]. Of note, when compared to non-ocular tissues, the retinal rate of detoxification was about 2-fold, 8.5-fold, 3.5-fold, and 4.5-fold greater than liver, heart, kidney, and brain, respectively (Figure 5E–H). From a teleological perspective, the high level of retinal GLO1 activity suggests an important protective role against AGE-derived damage in retina.

Several cautions are appropriate with regard to expectations of GLO1 overexpression, particularly in the lens. GLO1 may be found and function primarily in the epithelial layers where glucose is received from the aqueous or vitreous humors that nurture the lens. The epithelial layers comprise a minority of lens tissue; thus, the concentration of GLO1 might be significantly higher in the lens epithelium. This may explain glycation-related browning of lenses, particularly in the lens core, upon aging (Figure 2C). Given that the glyoxalase system declines in efficacy with age [189], enhancement of GLO1 activity might represent a therapeutic strategy to counteract the accumulation of these toxic compounds in the lens and retina.

The use of transcriptional modulators of GLO1 has been proposed as a potential intervention to support healthy aging and fight AGEs-related diseases. Metformin, an oral glucose-lowering agent for type 2 diabetes, increases GLO1 activity and was shown to lower circulating MG levels in patients with type 2 diabetes [202]. Candesartan, a synthetic drug that stimulates glyoxalase activity, reduced retinal acellular capillaries and attenuated inflammation and diabetic retinal vascular pathology [182]. In addition, dietary compounds including trans-resveratrol, fisetin, mangiferin, cyanidin, hesperetin, or sulforaphane possess stimulating GLO1-properties and lower MG and MG-derived adducts [152,153,154,155,156,203,204,205,206,207,208]. Proteomic analysis identified GLO1 as a protein differentially expressed in cells treated with sulforaphane [209]. Sulforaphane inhibited AGEs-derived pericyte damage [210] and delayed diabetes-induced retinal photoreceptor cell degeneration in streptozotocin-injected mice [211]. In addition, pyridoxamine, an MG scavenger that inhibits AGEs formation but also increases GLO1 activity [212], prevented the development of retinopathy in streptozotocin-induced diabetic rats [213]. There is a growing interest in drug discovery and high-throughput screening systems will identify novel regulators and small molecules with GLO1-stimulating properties [214], opening a possibility to treat DR. Clearly, further research is required to define nutritional and pharmacological approaches to overcome the progression of DR associated with glycative stress.

## 6. Conclusions

Currently there is no cure for DR. Modern therapies include anti-angiogenic strategies or surgery for retinal detachment, which cannot restore vision but only ameliorate further deterioration of the retina. A vast amount of literature stresses the pathogenic role of glycation and AGEs in the molecular basis of this eye-related disease. Based on this literature, lowering AGEs might be a potential therapeutic strategy against DR. Despite enormous effort to date, no AGEs inhibitors have reached clinical use. Exploiting the glyoxalase system and the discovery of compounds that enhance this detoxifying activity represent a therapeutic alternative to fight glycation-derived damage, under diabetic and non-diabetic conditions. As documented here, glyoxalase activity is highest in the retina, suggesting a significant role in eye physiology. The capacity of this detoxifying route is thought to decline with age. We propose that targeting the glyoxalase system through nutritional or pharmacological enhancers might be an alternative to fight this sight-threatening diabetic complication, aiming to reduce the economic burden caused by DR.

## Figures and Tables

**Figure 1 antioxidants-09-01062-f001:**
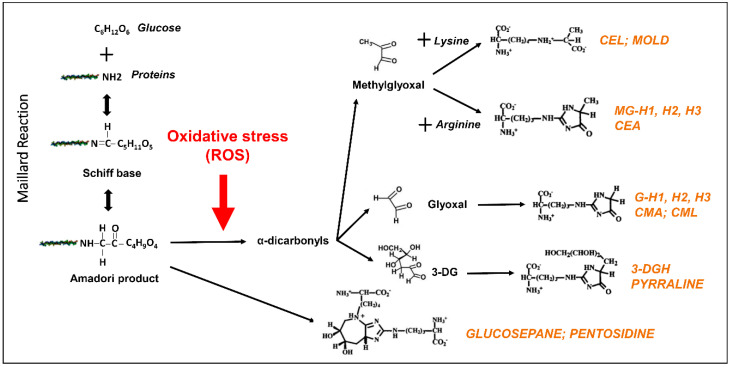
Production of advanced glycation end-products (AGEs) is accelerated under oxidative stress. AGEs are derived from sugars or dicarbonyls generated from carbohydrate metabolism through different chemical routes including the Maillard reaction. The excess of reactive oxidative species (ROS) promotes the production of different AGEs from dicarbonyls (methylglyoxal, glyoxal, or 3-deoxyglucosone (3-DG). AGEs are highlighted in orange. CML: Nε-(carboxymethyl)-lysine; CMA: Nε-(carboxymethyl)-arginine; 3-DG: 3-deoxyglucosone; 3-DGH: GH-1,2,3: Glyoxal-derived hydroimidazolone; MGH-1,2,3: Methylglyoxal-derived hydroimidazolone; CEL: Nε-(carboxyethyl)-lysine; CEA: Nε-(carboxyethyl)-arginine; MOLD: Methylglyoxal-derived lysine dimer.

**Figure 2 antioxidants-09-01062-f002:**
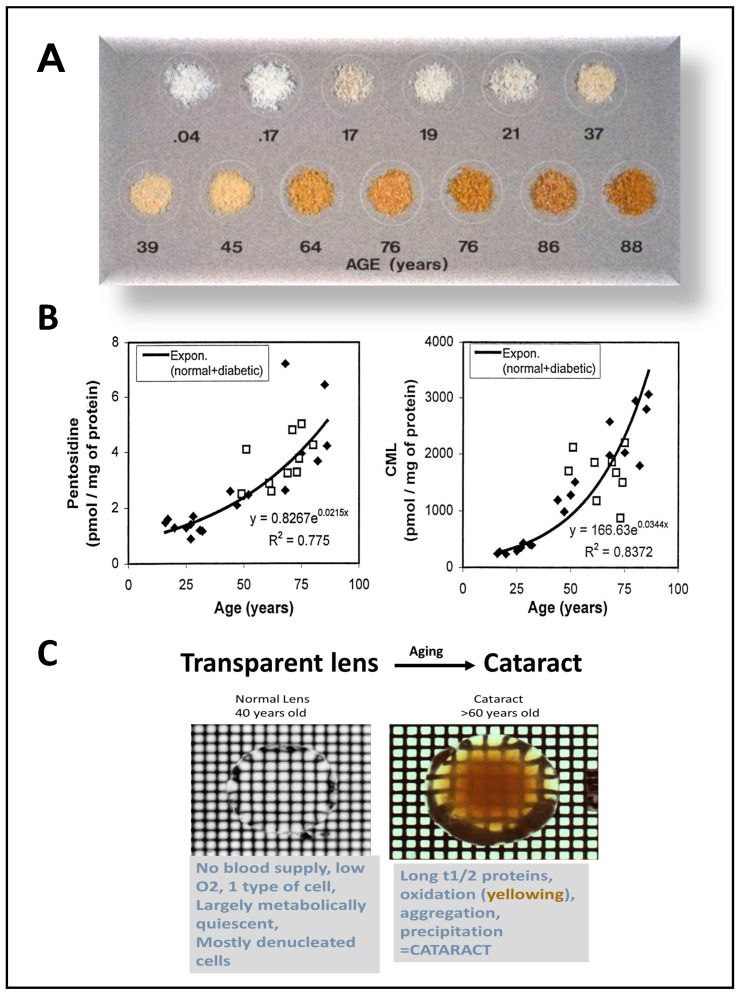
Aging promotes the accumulation of AGEs throughout the body. (**A**) Age-dependent changes in costal cartilage isolated at autopsy from donors of various ages (reprinted with permission of Dr. Baynes) [44]. (**B**) Level of pentosidine (left) and Nε-(carboxymethyl)-lysine (CML) (right) in lens crystallins from diabetic (■) and non-diabetic (♦) subjects as a function of age (reprinted with permission of Dr. Monnier) [46]. (**C**) Transillumination of isolated lenses: Normal lens from young donor (left) and cataractous lens from older donor (right). Lenses were placed in a culture dish that had a grid etched in its bottom surface. The dish was placed on the stage of a dissecting microscope and viewed with transmitted light. Note the degree of yellowing and opacity in the cataractous lens.

**Figure 3 antioxidants-09-01062-f003:**
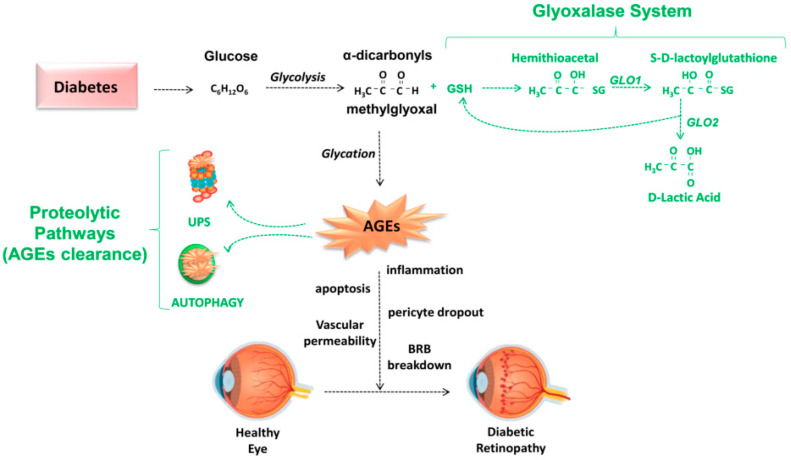
Schematic overview of detoxifying pathways against AGEs-derived damage in diabetic retinopathy (DR). Hyperglycemia-associated diabetes involves the abnormal formation of highly reactive α-dicarbonyls such as methylglyoxal which leads to accelerated AGEs formation. The glyoxalase system is a protective mechanism that slows down the synthesis of AGEs by limiting formation of dicarbonyls. Once formed, AGEs can be cleared by two proteolytic pathways: The ubiquitin-proteasome (UPS) system and autophagy. These protective mechanisms (highlighted in green) decline under diabetic conditions and with age. AGEs are pathologic features in the early stages of DR, impacting the function of neuroglial and vascular cells. AGEs-derived damage results in cellular and tissue dysfunction contributing to the onset of DR. BRB: Blood–retina barrier; GLO1: Glyoxalase 1; GLO2: Glyoxalase 2; GSH: Glutathione.

**Figure 4 antioxidants-09-01062-f004:**
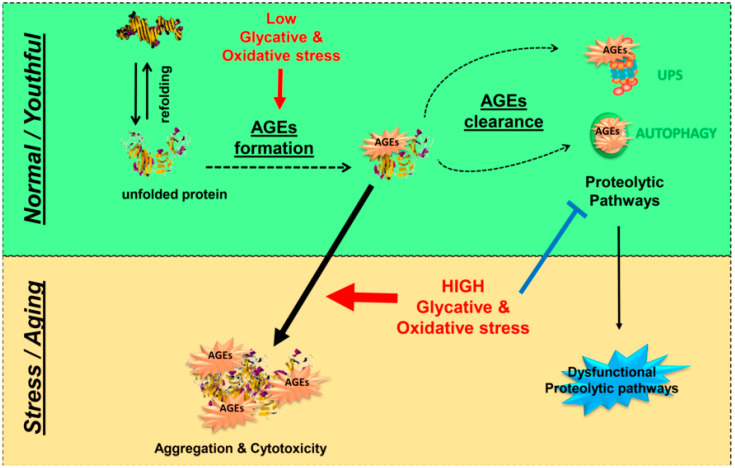
Proteolytic pathways are the last line of defense against AGE-derived proteotoxicity. Under homeostatic conditions (green box), different proteolytic pathways (UPS and autophagy) act to avoid the accumulation of toxic AGEs. Under high levels of glycative and oxidative stress and/or aging (orange box), the production of AGEs is boosted and tissue fitness is compromised as result of glycation-derived protein aggregation and cytotoxicity. Proteolytic capacities are insufficient to lower AGEs levels because of age-related changes in UPS and autophagy along with the inhibitory effect of glycative stress on proteolytic function.

**Figure 5 antioxidants-09-01062-f005:**
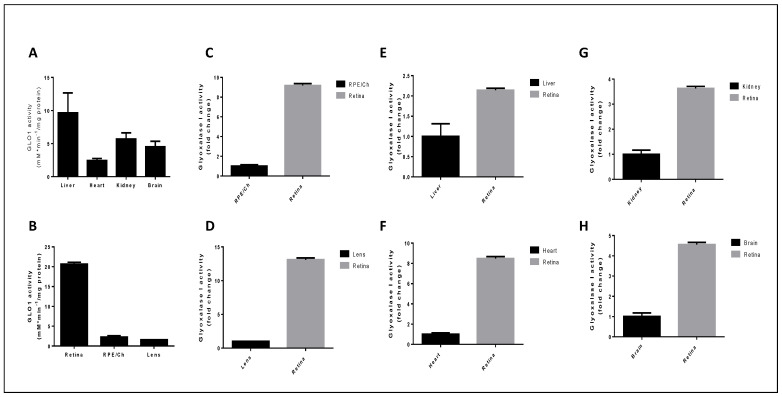
Evaluation of glyoxalase system activity in ocular and non-ocular tissues. Glyoxalase I activity was determined spectrophotometrically using 1 mL quartz cuvettes by following the initial rate of formation of S-D-lactoylglutathione. The assay mixture containing the glycating reagent MG and reduced GSH was equilibrated at room temperature for 10 min, to ensure hemithioacetal formation. The reaction was initiated by the addition of 20 μg of cytosolic extract and the A240 was monitored immediately and over the course of 5 min. The reaction rate was determined by following the increase in absorbance at 240 nm for which Δε240 = 2.86 mM^−1^ cm^−1^. (A,B) Glyoxalase I activity was assayed in (**A**) non-ocular tissues and (**B**) ocular tissues and activity was expressed as milliunits per milligram of protein where one unit of GLO1 activity was the amount of enzyme which catalyzes the formation of 1 μmol S-D-lactoylglutathione per min under assay conditions. (**C**–**H**) Retinal glyoxalase I activity was compared to (**C**) RPE/choroid, (**D**) lens, (**E**) liver, (**F**) heart, (**G**) kidney, and (**H**) brain. Fold change was calculated relative to each tissue and values represent the mean ± standard error of the mean of 4 independent experiments from the GLO1 activity assay.

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
