# Peer review of "Glyoxalase System as a Therapeutic Target against Diabetic Retinopathy"

_antioxidants, 2020, doi:10.3390/antiox9111062_

Round 1

Reviewer 1 Report

The Review of Aragonès and colleagues reported a for the first time the relations between oxidative stress, formation of AGEs, and aging in several tissues of the eye. They summarize mechanisms of AGEs generation and anti-AGEs detoxifying systems; whit particular attention to the potential of the glyoxalase system in the retina in the prevention of AGEs-associated damage linked to DR.

The Authors reported a comparative analysis of glyoxalase activity in different tissues from wild type mice, supporting a major role for the glyoxalase system in detoxification of AGEs in the retina, and present the manipulation of this system as a therapeutic strategy to prevent the onset of DR.

The review is well organized, well written and reported the current data present in literature concerning this topic. Moreover, the presence of the schemes/figures helps to better understand the topic.

Only minor revision:

Lane 17 à loop add an “s” : loops;

Lane 45 à correct the sentence: “this 45 imbalance” with These 45 imbalances;

Lane 135 à occurs remove the “s”: occur;

Lane 295 à dimer add an “s” : dimers.

Author Response

As the reviewer suggested, we have corrected these mistakes.

Reviewer 2 Report

In thsi Review, Aragonès and colleagues describe the detrimental role of AGEs in diabetic retinopathy (DR) and focus on the importance of the retina glyoxalase (GLO) system in preventing AGEs-associated damage in DR. Moreover, since  in mice a major role for the glyoxalase system in detoxification of AGEs in the retina is observed, they suggest that the manipulation of GLO system may be a potential therapeutic strategy to prevent the onset of DR. 

The Review is very interesting, complete, well written and organized. 

English just needs a minor revision.

I have just a few suggestions to make:

i) Lines 272 and 304: I would add a recent review here (PMID: 29385039)

ii) Line 406: I would add some recent and interesting artciles demonstrating Glo1 regulation by Nrf2 (PMID: 29170092; PMID: 29511711; PMID: 32756399)

iii) Line 413: I would add here also that NF-kB directly can downregulate Glo1 activity (PMID: 25841781)

Author Response

 As the reviewer suggested, we have added these recent and interesting articles in the manuscript (references #98, #149, #150, #151 and #158).

Reviewer 3 Report

This is a well-written and organized review on the glyoxalase on its potential association with DR and it was suggested as a therapeutic target for DR. It would be better if the author could include some discussion on which compounds/treatments can be used to modulate glyoxalase to act as potential therapeutics for DR.

Author Response

Thank you for your comment. We have added a paragraph (page 14; from line 553 to 568) about the use of multiple compounds (natural or synthetics) as a modulators of glyoxalase system in order to prevent or alleviate glycation-derived damage with age-related diseases such as DR.